# Enhanced Mechanical Properties of Polylactic Acid/Poly(Butylene Adipate-co-Terephthalate) Modified with Maleic Anhydride

**DOI:** 10.3390/polym16040518

**Published:** 2024-02-14

**Authors:** Kibeom Nam, Sang Gu Kim, Do Young Kim, Dong Yun Lee

**Affiliations:** Department of Polymer Science and Engineering, Kyungpook National University, Daegu 41566, Republic of Korea; ska2918@naver.com (K.N.); sgkim@wschemical.co.kr (S.G.K.); ddyykk9655@gmail.com (D.Y.K.)

**Keywords:** biodegradable plastics, polylactic acid, poly(butylene adipate-co-terephthalate), maleic anhydride, grafting, compounding

## Abstract

An increase in plastic waste pollution and the strengthening of global environmental policies have heightened the need for research on biodegradable plastics. In this regard, polylactic acid (PLA) and poly(butylene adipate-co-terephthalate) (PBAT) are notable examples, serving as alternatives to traditional plastics. In this study, the compatibility and mechanical properties of PLA/PBAT blends were improved by the chemical grafting of maleic anhydride (MAH). In addition, qualitative analyses were conducted, dynamic mechanical properties were investigated, and the structure and mechanical characteristics of the blends were analyzed. With an increase in the MAH concentration, the grafting yield of the blends increased, and significantly improved the compatibility of the PLA/PBAT blend, with an ~2 and 2.9 times increase in the tensile strength and elongation at break, respectively. These findings indicate that the modified PLA/PBAT blend demonstrates potential for applications that require sustainable plastic materials, thereby contributing to the development of environmentally friendly alternatives in the plastics industry.

## 1. Introduction

The industrial revolution and the resulting increase in the use of plastics have led to an annual production of more than 250 million tons of plastic waste worldwide [1]. Although plastics exhibit advantages in terms of their processing and cost-effectiveness, their extensive use has resulted in significant environmental pollution due to their disposal in landfills and their incineration. This escalating environmental concern has led to stringent regulations, especially in the European Union, Japan, and the United States, aimed at curbing the use of single-use plastics [2,3,4,5]. Consequently, the demand for sustainable biomass-based plastic alternatives has increased. Biodegradable plastics, which completely decompose into water and carbon dioxide via their interaction with soil, have emerged as a potential solution to the plastic waste issue.

Thermoplastic starch (TPS) is increasingly recognized as a promising material for creating sustainable biodegradable plastics, particularly in the food-packaging sector [6]. Derived mainly from natural polysaccharides like corn, wheat, and potatoes, TPS is created through a gelatinization process involving starch and glycerol [7,8]. This process breaks down hydrogen bonds in starch molecules, preventing retrogradation and enhancing flexibility [9]. TPS is advantageous because it can be processed using conventional plastic equipment, making it suitable for both rigid and flexible packaging applications. However, TPS faces challenges like high hygroscopicity and poor mechanical performance, which limit its commercial viability.

Polybutylene succinate (PBS) is recognized for its mechanical properties and processability, making it an attractive biodegradable polymer for various applications including in the packaging, automotive, and biomedical fields [10]. Derived from butanediol and succinic acid, PBS stands out for its compatibility with both bio-based and petroleum-based resources, offering a sustainable alternative to traditional plastics [11]. Its blending with natural fibers or fillers, like cellulose nanofibers, enhances mechanical performance, although challenges exist in ensuring compatibility between hydrophobic PBS and hydrophilic natural fillers [12]. PBS, while offering numerous advantages as a biodegradable polymer, does have its limitations. One notable drawback is its brittleness compared with other polymers like PBAT, which can limit its applicability as a film material.

Polylactic acid (PLA) and poly(butylene adipate-co-terephthalate) (PBAT) are notable examples of biodegradable plastic materials. PLA, derived from renewable resources such as corn starch and cassava sugars, decomposes completely after use, exhibiting cost-effectiveness, impressive mechanical strength, and biocompatibility. However, its application scope is limited due to inherent drawbacks such as low fracture toughness, high brittleness, slow crystallization, and a narrow processing window, restricting its use in processes such as blown film extrusion and foam formation [13,14,15]. These limitations have been resolved by modifying PLA with compatibilizers such as plasticizers and reactive coupling agents, thereby enhancing its processability in the packaging industry [16,17,18,19].

As another widely recognized biodegradable plastic material derived from petroleum-based resources, PBAT is known for its high elongation at break and its good ductility, similar to those of thermoplastic elastomers [20]. Despite its advantages, PBAT exhibits disadvantages such as high cost, low stiffness, and weak tensile strength, limiting its practical applications. To resolve these issues, blending PBAT with PLA has been proposed, combining their complementary properties to create a material suitable for film packaging [21,22,23,24]. However, PLA and PBAT exhibit low miscibility, leading to deteriorated mechanical properties after blending. The application of casting or extrusion molding methods, typically used for PLA films, has been successfully adapted to blown molding processes [25,26,27,28]. In this study, the development of a biodegradable film modified with MAH enabled the use of the blown molding process, which was previously challenging due to the introduction of side chains and enhanced rheological properties. Notably, the improvement in rheological properties allowed us to produce thin films, a feat difficult to achieve with casting or extrusion molding methods. Additionally, the continuous process of blown molding offers the advantage of increased productivity, making this technology more practical and versatile for various applications. This adaptation expands the processing versatility of PLA-film manufacturing. To enhance the compatibility of these polymers, the use of compatibilizers and the addition of nanoparticles and inorganic materials have been explored [29,30,31]. In addition, studies have reported the incorporation of other polymers to create ternary blends [32,33,34], as well as the use of grafting methods using MAH [35,36,37]. Notably, research involving the grafting of MAH, specifically using PLA-g-MAH as a masterbatch to analyze the variation in properties, is progressing. However, this approach is not convenient as an additional processing step is required after producing PLA-g-MAH. Furthermore, limited studies on the comparison of the mechanical properties of resins using a masterbatch with PLA/PBAT/MAH produced in a single process are available.

In the context of these developments, plasticized PLA/PBAT blown films were prepared using suitable plasticizers to enhance their tear resistance. Moreover, the thermal and mechanical properties, as well as tear resistance and morphology, of the plasticized PLA/PBAT blends and films, were comprehensively evaluated. The goal of this study was to identify the most effective plasticizer for PLA and investigate the effects of various plasticizers and concentrations on the material properties. The results indicated that the PBAT blown films containing PLA plasticized with adipic acid ester, prepared by a blown process, exhibited significantly improved mechanical properties and tear resistance in comparison with the films without plasticizers.

## 2. Materials and Methods

### 2.1. Materials

As the biodegradable plastic material PLA (Ingeo™ Biopolymer 4032D, NatureWorks LLC, Minnetonka MN, USA), and PBAT (EnPol PBG7070, Lotte Chemical Co., Seoul, Republic of Korea) were purchased. As the initiators dicumyl peroxide (DCP, Sigma-Aldrich, St. Louis, MO, USA) was purchased, and grafting materials MAH (Sigma-Aldrich, St. Louis, MO, USA) was purchased. Figure 1 shows the molecular structures.

### 2.2. Preparation of PLA or PBAT Composites and Blend Samples

PLA or PBAT was simultaneously modified into a masterbatch using an initiator and grafting materials after drying in a vacuum oven at 50 °C for 24 h to remove moisture. By using the MAH-modified masterbatch, PLA-g-MAH/PBAT and PLA/PBAT-g-MAH composites were fabricated. These processes were conducted by melt-mixing methods using a Plasti-Corder Lab-Station with a melt mixer (W50 EHT, Brabender, Duisburg, Germany) equipped with a counter-rotating twin-screw compounder with a bowl volume of 55 cm^3^ and roller blades. A barrel temperature of 180 ± 5 °C with a rotation speed of 50 rpm and a total residence time of 10 min were utilized. Table 1 summarizes the compositions of the PLA and PBAT composites.

The prepared compounds were cut into small pieces and shaped into 140 mm × 120 mm rectangles with a thickness of 1 mm using a hydraulic laboratory press (Model 3851, Fred S. Carver Inc., Menomonee Falls, WI, USA). Preheating at 180 °C for 4 min, followed by molding at a pressure of 2000 psi for 2 min and cooling, afforded the desired modified PLA materials.

### 2.3. Instrumentation and Equipment

Titration analysis was performed to determine the grafting yield of the PLA-g-MAH and PBAT-g-MAH samples. The grafted samples were dissolved in 50 mL of chloroform, followed by the addition of 1 mL of 0.1 M HCl and stirred for 30 min. After stirring, the samples were precipitated in excess methanol and then filtered. The filtered samples were dried at 80 °C for 24 h. Once the samples were dried, the quantified samples were redissolved in chloroform. To this solution, 1 mL of a 1% phenolphthalein solution diluted in ethanol was added, followed by titration with 0.03 N KOH. The grafting yield of MAH was calculated using the following formulae:$$Acid\;Number\left(mg\;KOH/g\right)=\frac{mL\;KOH\times N\;KOH\times 56.1}{g\;Polymer}$$
$$MAH\;Grafting\;Yield\left(\%\right)=\frac{Acid\;Number\times 98.06}{2\times 561}$$

The morphologies of the film samples were investigated using field-emission scanning electron microscopy (FE-SEM; SU8220, Hitachi, Tokyo, Japan) at an accelerating voltage of 5 kV. The samples were cryogenically fractured in liquid nitrogen and sputtered with Pt for 60 s.

Dynamic mechanical analysis (DMA) measurements of the samples were conducted using a dynamic mechanical analyzer (N535, Perkin-Elmer, Boston, MA, USA) at temperatures from −70 °C to 180 °C, a heating rate of 5 °C/min, and a frequency of 1 Hz, under nitrogen. Samples with dimensions of 30 mm × 10 mm × 1 mm were prepared using a hydraulic laboratory press.

The blown film was fabricated using a blown film machine (SJ45-MGF600, Seojin Biotec, Incheon, Republic of Korea) with a 45 mm single screw. The annular die diameter was 80 mm and the die gap was 1.2 mm. The barrel temperature profile was set at 150/165/170/140 °C and the head and die temperatures were 165 and 170 °C, respectively. The blowing conditions for the film were fixed at a blow-up ratio of 2.8 and a film thickness of 30 µm.

The mechanical properties of the samples were evaluated using a universal testing machine (UTM; LR5K Plus, Lloyd Instruments, West Sussex, UK) equipped with a 500 N load cell and an Elmendorf tear tester (ElmaTear 855, James H. Heal & Co. Ltd., Halifax, UK). Furthermore, the tensile properties were analyzed according to the ASTM D882 standards [38]. Samples were prepared with a thickness of 1 mm using a hydraulic laboratory press. These were then precisely cut into a dog-bone shape, characterized by 10 mm wide ends and a 4 mm wide neck, the latter being the region where tensile stress is predominantly applied. The prepared samples were then affixed securely in the grips of a UTM. A tensile test was conducted at a consistent extension rate of 25 mm/s, pulling the sample vertically to evaluate its mechanical strength under controlled conditions.

For tear-strength analysis, a rectangular shape measuring 63 mm × 76 mm, was also employed from blown film. In cases involving textiles, a modified rectangular sample with additional height at the ends was used to reduce edge unraveling. These tests were performed using an Elmendorf tear tester, following the guidelines set by ASTM D1922 [39]. For tear-resistance tests, five overlapped samples from each group were analyzed in the machine direction (MD) and transverse direction (TD).

## 3. Results and Discussion

### 3.1. Spectrophotometric Analysis and Grafting Yield

The free radical grafting of MAH onto PLA or PBAT involves several distinct reaction mechanisms, as illustrated in Figure 2. The primary mechanism, as suggested by previous research, entails hydrogen abstraction from the α-carbon of the adipate segment [40]. This process is facilitated by the free radical stabilization effects of the carbonyl group, leading to the formation of a chemical bond between the maleic anhydride group and the PBAT chain, ultimately yielding a succinic anhydride moiety as a pendant group on the PBAT backbone. This graft copolymerization reaction is further characterized by a β-scission reaction, which splits the chain, resulting in the formation of a succinic anhydride end group. Moreover, the generated anhydride free radical is capable of abstracting additional hydrogens or undergoing coupling (i.e., recombination) with other free radical entities.

Another mechanism proposed for the grafting of maleic anhydride onto polyesters involves chain-end grafting. This process starts with the abstraction of hydrogen from the α-carbon, followed by β-scission leading to the creation of vinylidene and macroradical chain ends [41]. These macroradical ends are then able to interact with MAH groups, following various pathways to couple with additional free radical moieties. It is important to highlight that β-scission tends to be favored during melt processing, whereas in solution-phase radical reactions, the coupling of radicals prior to scission becomes the predominant mechanism [42].

Furthermore, the grafting reaction, conducted at an extrusion temperature of 185 °C, is not conducive to the homopolymerization of MAH units, ensuring the specificity of the grafting process [43,44,45]. The grafting yield was confirmed through FTIR spectroscopy and titration methods, while the grafting efficiency was determined by comparing the actual grafted MAH content with the initial amount of MAH used.

Figure 3 shows the FTIR spectra of PLA-g-MAH, PBAT-g-MAH, and (PLA/PBAT)-g-MAH. The FTIR spectrum of MAH revealed peaks at 3090 cm^−1^ and 1600 cm^−1^, attributed to the stretching vibrations of C–H and C=C groups, respectively. Peaks at 1853 cm^−1^ and 1780 cm^−1^ corresponded to the symmetric and asymmetric stretching vibrations of C=O, respectively. The IR spectrum of PLA revealed peaks of C–H and C=O groups at 2926 cm^−1^ and 1750 cm^−1^, respectively, with peaks of C–O–C groups observed at ~1190 cm^−1^, 1130 cm^−1^, and 1093 cm^−1^. In the IR spectrum of PBAT, peaks of C–H and C=O groups were observed at 2960 cm^−1^ and 1710 cm^−1^, respectively, and peaks of C–O–C and CH_2_ groups were observed at 1269 cm^−1^ and 727 cm^−1^, respectively. The FTIR spectrum of PLA-g-MAH (Figure 3a) revealed PLA peaks with no MAH peaks at 3090 cm^−1^ or 1853 cm^−1^, indicating the absence of unreacted MAH [46,47]. Figure 3b shows that the FTIR spectrum of PBAT-g-MAH, also lacking MAH peaks, was similar to that of PLA-g-MAH.

Figure 4 shows the relationship between the grafting yield of polymer blends and the concentration of MAH, demonstrating a clear positive relationship where the grafting yield increased with the elevation in MAH content. This observation emphatically supports the role of MAH as an efficacious grafting agent, facilitating significant chemical modifications to the polymer chains within the blends. Moreover, a comparative assessment of the grafting yields for PBAT and PLA, under uniform MAH concentrations, revealed a pronounced higher grafting efficiency in PBAT compared with PLA. This variance can be attributed to the distinct reactivity profiles of the polymers towards MAH, particularly the enhanced ability for hydrogen abstraction from the α-carbons in the polymer chains observed in PBAT. The structural characteristics of PBAT, influenced by its chemical composition and configuration, may confer a higher affinity towards MAH grafting, culminating in superior yield outcomes. These insights are consistent with findings from previous research on the grafting reactions involving PLA-g-MAH and PBAT-g-MAH [45], collectively reinforcing the premise that the intrinsic structural differences between PLA and PBAT markedly affect their reactivity with MAH. The observed disparities in grafting yield and efficiency not only shed light on the polymers’ chemical attributes but also underscore the criticality of meticulously selecting grafting conditions and MAH concentrations to refine the properties of the resultant blends effectively.

Table 2 shows the grafting yield of the different blends. The blend identified as PBAT-g-MAH-3 emerged with the highest grafting yield, registering at 1.20%, a figure that notably surpassed the yield of PLA-g-MAH-1, which was recorded at the lower end of the spectrum at 0.51%. These variations in grafting efficiency provide evidence of the influences of MAH concentration on the efficacy of the grafting reaction.

Moreover, the data reveal an increase in grafting yield from PBAT-g-MAH-1 to PBAT-g-MAH-3, with PBAT-g-MAH-1 showing a significantly lower yield than its subsequent iterations. This trend underscores the integral role of MAH concentration in optimizing the grafting efficiency. The reduced yield noted in PBAT-g-MAH-1 can be ascribed to the limited availability of MAH for grafting, a situation induced by the radical generation process initiated by the initiator. In conditions where MAH is scant, the likelihood of crosslinking within the PBAT matrix escalates, potentially impeding the successful grafting of MAH onto the polymer chain, as evidenced by the referenced studies [45,48].

This analysis not only elucidates the impact of MAH concentration on grafting yields, but also highlights the necessity for careful control over MAH content to ensure optimal grafting outcomes, thereby enhancing the material properties of the resultant polymer blends.

### 3.2. Morphology Characterization

To observe the structural changes that occurred with the addition of MAH, the fractured surfaces of the PLA/PBAT composites with different MAH concentrations were examined (Figure 5). Typically, in immiscible composite materials, a sea-island morphology is observed, where one polymer forms domains, while the other acts as the matrix. A sea-island structure was clearly observed for PLA/PBAT (Figure 5), contributing to the degradation of the mechanical properties of the immiscible polymer composites [21,49]. However, PLA-g-MAH/PBAT exhibited almost no sea-island morphology, regardless of the MAH concentration, suggesting increased compatibility between PLA and PBAT as a result of grafting with MAH (Figure 5a) [38,50,51]. Similarly, PLA/PBAT-g-MAH and (PLA/PBAT)-g-MAH also exhibited a nearly absent sea-island structure on their fractured surfaces (Figure 5b,c), indicating improved miscibility in these composites. Furthermore, the presence of unreacted MAH molecules in PLA-g-MAH may serve as a plasticizing agent for the blends, thereby influencing their elongation at break [52].

### 3.3. Rheological and Mechanical Properties

Figure 6 presents the tan δ graph for MAH-modified PLA/PBAT composites across a range of temperatures. Figure 6(a-1), (b-1) and (c-1) indicate the glass transition temperatures (T_g_) of PBAT, while Figure 6(a-2), (b-2) and (c-2) indicate the glass transition temperatures (T_g_) of PLA, respectively. This illustrates the relationship between the tan δ values and the thermal transitions of the polymers, highlighting the distinct T_g_ of each component within the composite.

Previous studies have established a correlation between rheological moduli and properties of polymers such as the glass transition temperature, chain entanglement, molecular weight, and cross-linking density [53,54,55]. Additionally, the mechanical behavior of polymers has been shown to be significantly influenced by several parameters, including molecular weight and its distribution, chain structure, cross-linking density, crystallinity, and the overall composition of the polymer [56,57,58,59]. Consequently, modifications to any of these variables are likely to induce substantial changes in the flow characteristics of the polymer. Such alterations can be effectively elucidated through the study of the polymer’s rheological behavior [60,61].

The curves of tan δ were meticulously generated for both pristine PLA and PBAT, and their MAH-grafted counterparts to evaluate the synergistic effects on the rheological properties of PLA and PBAT. This analysis was pivotal in understanding the impact of MAH modification on the composite materials, providing insights into the enhanced interactions between PLA and PBAT when modified with MAH.

The T_g_ values of PBAT and PLA in the unmodified PLA/PBAT composite were −26.7 °C and 61.3 °C, respectively. Generally, the compatibility of two polymers in a composite has been inferred from the variation in their T_g_ values; similar T_g_ values indicate increased compatibility [29,62,63]. For PLA-g-MAH/PBAT, the T_g_ values of PBAT and PLA decreased (Figure 6a); specifically, the T_g_ values of PBAT and PLA decreased to −31.5 °C and 58.7 °C (PLA-g-MAH/PBAT_1.5), respectively. Conversely, in the case of the PLA/PBAT-g-MAH and (PLA/PBAT)-g-MAH composites, the T_g_ of PBAT exhibited minor changes, while the T_g_ of PLA decreased. This result suggested that during the melt-mixing process, MAH preferentially reacted with PLA or PBAT, influencing their respective grafting. Specifically, the T_g_ of PLA in PLA/PBAT-g-MAH_1.0 decreased to 58.5 °C, while that of (PLA/PBAT)-g-MAH decreased to 58.6 °C, indicating improved compatibility due to MAH addition. Table 3 summarizes the changes in the T_g_ values.

Figure 7 and Appendix A show the tensile strength and elongation at break of the PLA/PBAT composites as a function of the MAH concentration. The MAH concentration did not significantly alter the tensile modulus, tensile strength, or elongation, which was consistent with the results reported previously [64]. However, compared with the composites with 1.0 and 1.5 phr of MAH, PLA/PBAT-g-MAH_0.5 exhibited a substantial decrease in the elongation at break, likely attributed to the β-scission reactions in the PLA main chain caused by an excess of DCP relative to MAH [65,66,67]. Among samples with the same MAH concentration, (PLA/PBAT)-g-MAH exhibited marginally higher mechanical properties than those of PLA-g-MAH/PBAT and PLA/PBAT-g-MAH. This result was attributed to the crosslinking between PLA and PBAT in the presence of active radicals [47,68,69]. Table 4 summarizes the modulus, tensile strength, and elongation at break of each sample. (PLA/PBAT)-g-MAH_1.0 exhibited the highest modulus and tensile strength of 679.3 and 27.7 MPa, respectively. Compared with those of the unmodified PLA/PBAT composite, the mechanical properties reflect an ~2.8 times increase in the tensile modulus from 243.1 MPa and a doubling of the tensile strength from 13.9 MPa. The elongation increased approximately 2.9 times from 132.7% to 380.9%. These results were consistent with the earlier analyzed structural and rheological characteristics, suggesting that grafting with MAH affected the compatibility and intermolecular interactions between PLA and PBAT [70,71].

## 4. Conclusions

In this study, the compatibility, and mechanical properties of a widely used biodegradable plastic material (PLA/PBAT) were enhanced by the incorporation and grafting of MAH. In addition, the structural characteristics, titration, rheological behavior, and thermal properties of the MAH-grafted PLA and PBAT samples were comprehensively evaluated. MAH tended to form a more pronounced branched structure in PBAT than in PLA.

FTIR analysis confirmed the grafting of MAH on PLA-g-MAH and PBAT-g-MAH. The grafting yield of MAH was quantified using titration, revealing an increase in the grafting yield with an increase in the MAH concentration. The grafting yield of PBAT was notably greater than that of PLA. Structural analysis revealed that unmodified samples exhibited a sea-island morphology, which became less distinct with the addition of MAH, indicative of improved miscibility. DMA demonstrated that the addition of MAH reduced the disparity between the T_g_ of PLA and PBAT, suggesting enhanced compatibility. In terms of mechanical properties, the addition of MAH led to an approximately 2.8-fold increase in the tensile modulus, a doubling of tensile strength, and a 2.9-fold increase in the elongation at break of the PLA/PBAT composite.

In conclusion, the addition of MAH significantly enhanced the compatibility of the PLA/PBAT composite and markedly improved its modulus, tensile strength, and elongation. These results highlight the potential of the modified PLA/PBAT blend for various applications within the sustainable plastics industry and demonstrate the capabilities of high-performance biodegradable plastic materials for manufacturing disposable items in medical and food industries, aligning with the global shift towards sustainable solutions. Beyond these, the composite’s enhanced features could find applications in agricultural sectors, particularly in the production of biodegradable mulch films and plant containers.

## Figures and Tables

**Figure 1 polymers-16-00518-f001:**
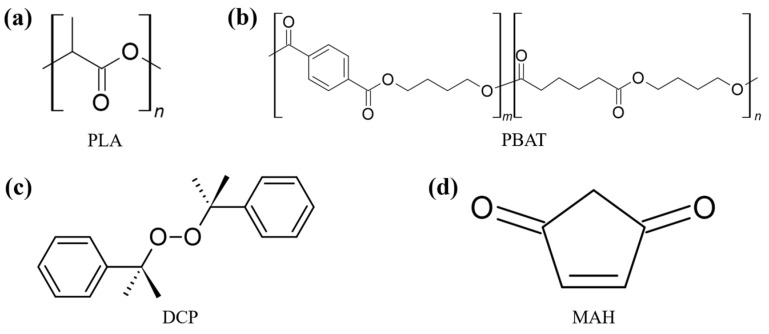
Chemical structures of (**a**) polylactic acid (PLA), (**b**) poly(butylene adipate-co-terephthalate) (PBAT), (**c**) dicumyl peroxide (DCP), and (**d**) maleic anhydride (MAH).

**Figure 2 polymers-16-00518-f002:**
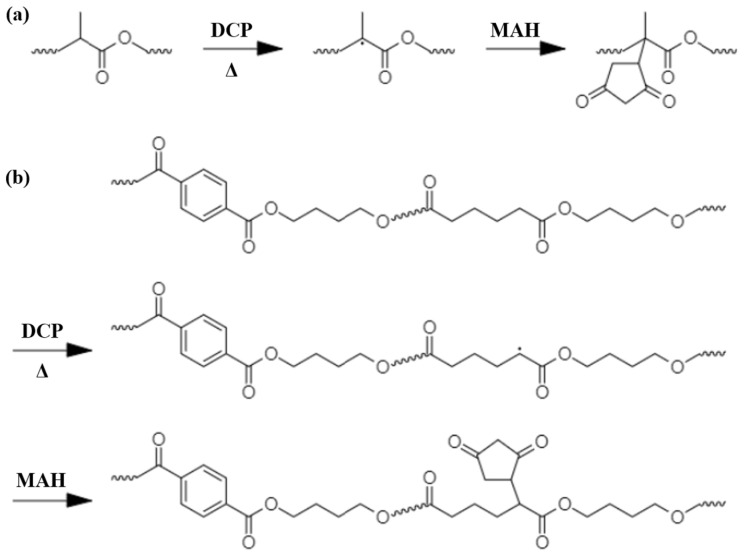
Proposed reaction mechanism of the grafting of maleic anhydride (MAH) on (**a**) polylactic acid (PLA) and (**b**) poly(butylene adipate-*co*-terephthalate) (PBAT).

**Figure 3 polymers-16-00518-f003:**
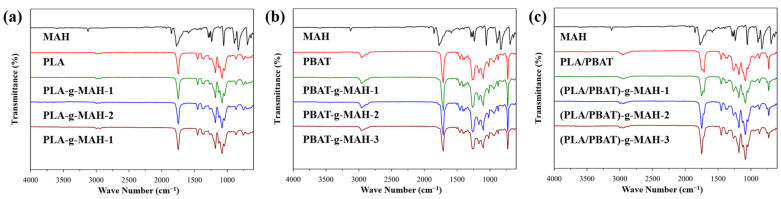
FTIR spectra of (**a**) PLA-g-MAH, (**b**) PBAT-g-MAH, and (**c**) (PLA/PBAT)-g-MAH.

**Figure 4 polymers-16-00518-f004:**
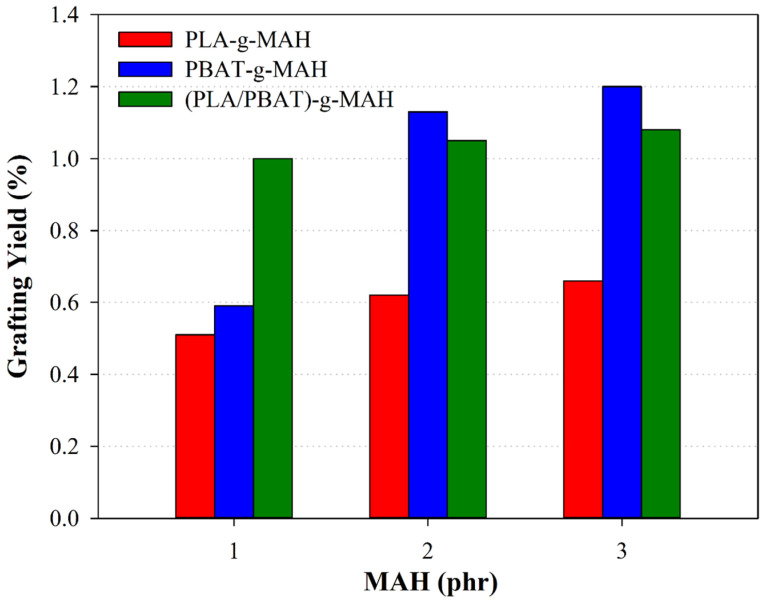
Grafting yields as a function of the MAH content.

**Figure 5 polymers-16-00518-f005:**
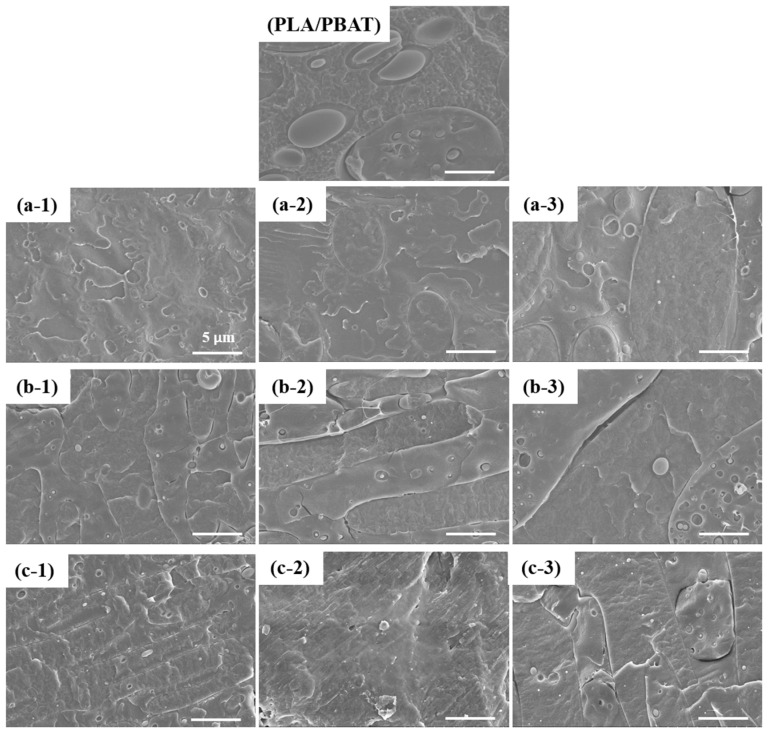
SEM images of (**a-1**) PLA-g-MAH/PBAT_0.5, (**a-2**) PLA-g-MAH/PBAT_1.0, (**a-3**) PLA-g-MAH/PBAT_1.5, (**b-1**) PLA/PBAT-g-MAH_0.5, (**b-2**) PLA/PBAT-g-MAH_1.0, (**b-3**) PLA/PBAT-g-MAH_1.5, (**c-1**) (PLA/PBAT)-g-MAH_0.5, (**c-2**) (PLA/PBAT)-g-MAH_1.0, (**c-3**) (PLA/PBAT)-g-MAH_1.5.

**Figure 6 polymers-16-00518-f006:**
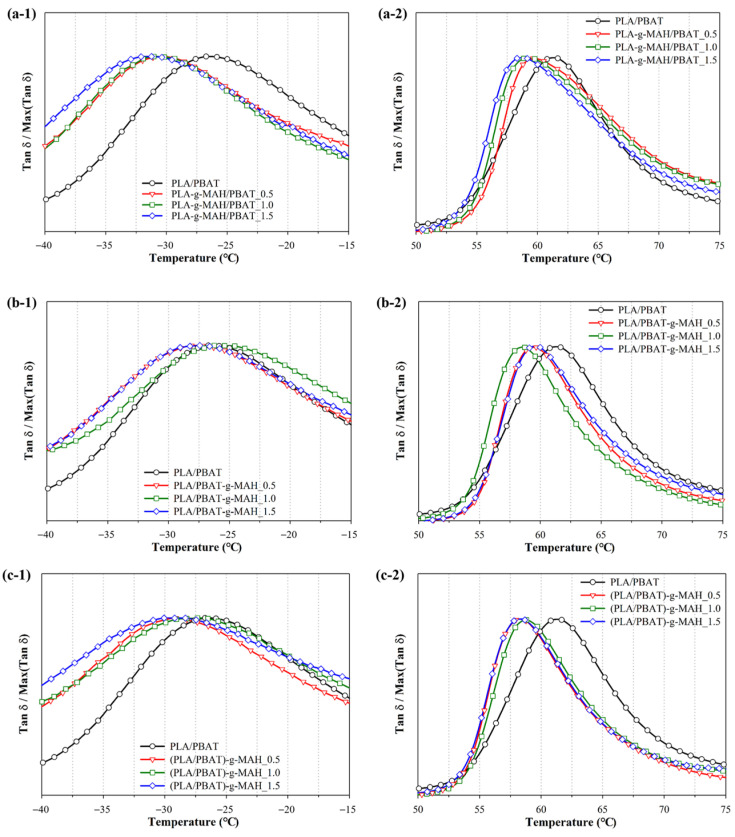
Variation of tan δ with temperature for the PLA/PBAT blends with various MAH concentrations: tan δ of (**a-1**) PLA-g-MAH/PBAT, (**b-1**) PLA/PBAT-g-MAH, and (**c-1**) (PLA/PBAT)-g-MAH in the range from −40 to −15 °C, and tan δ of (**a-2**) PLA-g-MAH/PBAT, (**b-2**) PLA/PBAT-g-MAH, and (**c-2**) (PLA/PBAT)-g-MAH in the range from 50 to 75 °C.

**Figure 7 polymers-16-00518-f007:**
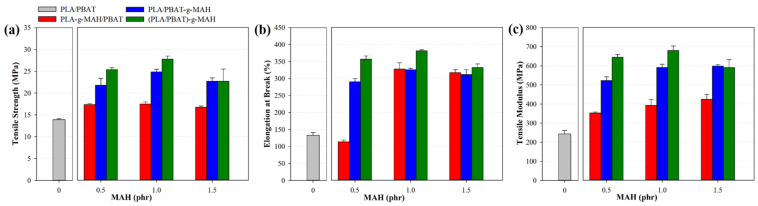
(**a**) Tensile strength, (**b**) elongation at break, and (**c**) tensile modulus of PLA/PBAT blends with various MAH concentrations.

**Table 1 polymers-16-00518-t001:** Composition of PLA/PBAT blends with various MAH contents.

**(a)**				
**Description**	**PLA (wt%)**	**PBAT (wt%)**	**DCP (phr)**	**MAH (phr)**
PLA	100			
PBAT	-	100		
PLA/PBAT	50	50		
DCP_0.5	50	50	0.5	
MAH_0.5			-	0.5
MAH_1.0			-	1.0
MAH_1.5			-	1.5
PLA-g-MAH-1	100	-	1	1
PLA-g-MAH-2				2
PLA-g-MAH-3				3
PBAT-g-MAH-1	-	100		1
PBAT-g-MAH-2				2
PBAT-g-MAH-3				3
(PLA/PBAT)-g-MAH-1	50	50		1
(PLA/PBAT)-g-MAH-2				2
(PLA/PBAT)-g-MAH-3				3
**(b)**				
**Description**	**Component 1 (50 wt%)**		**Component 2 (50 wt%)**	
PLA-g-MAH/PBAT_0.5	PLA-g-MAH-1		Neat PBAT	
PLA-g-MAH/PBAT_1.0	PLA-g-MAH-2			
PLA-g-MAH/PBAT_1.5	PLA-g-MAH-3			
PLA/PBAT-g-MAH_0.5	Neat PLA		PBAT-g-MAH-1	
PLA/PBAT-g-MAH_1.0			PBAT-g-MAH-2	
PLA/PBAT-g-MAH_1.5			PBAT-g-MAH-3	
**(c)**				
**Description**	**PLA (wt%)**	**PBAT (wt%)**	**DCP (phr)**	**MAH (phr)**
(PLA/PBAT)-g-MAH_0.5	50	50	0.5	0.5
(PLA/PBAT)-g-MAH_1.0				1.0
(PLA/PBAT)-g-MAH_1.5				1.5

**Table 2 polymers-16-00518-t002:** Grafting yield of PLA/PBAT blends with various MAH concentrations.

Description	DCP (phr)	MAH (phr)	Grafting Yield (%)
PLA-g-MAH-1	1	1	0.51
PLA-g-MAH-2		2	0.62
PLA-g-MAH-3		3	0.66
PBAT-g-MAH-1		1	0.59
PBAT-g-MAH-2		2	1.1
PBAT-g-MAH-3		3	1.20
(PLA/PBAT)-g-MAH-1		1	1.00
(PLA/PBAT)-g-MAH-2		2	1.05
(PLA/PBAT)-g-MAH-3		3	1.08

**Table 3 polymers-16-00518-t003:** Tan δ results of PLA/PBAT blends with various MAH concentrations.

Description	PBAT T_g_ (°C)	PLA T_g_ (°C)	PLA T_g_–PBAT T_g_ (°C)
PLA/PBAT	−26.7	61.3	88.0
PLA-g-MAH_0.5	−30.5	59.6	90.1
PLA-g-MAH_1.0	−31.0	59.0	90.0
PLA-g-MAH_1.5	−31.5	58.7	89.7
PBAT-g-MAH_0.5	−27.8	59.4	87.2
PBAT-g-MAH_1.0	−25.8	58.5	84.3
PBAT-g-MAH_1.5	−27.5	59.5	87.1
(PLA/PBAT)-g-MAH_0.5	−29.1	58.3	87.4
(PLA/PBAT)-g-MAH_1.0	−27.1	58.6	85.7
(PLA/PBAT)-g-MAH_1.5	−29.5	58.3	87.8

**Table 4 polymers-16-00518-t004:** Mechanical properties of PLA/PBAT blends with various MAH concentrations.

Description	Tensile Modulus (MPa)	Tensile Strength (MPa)	Elongation at Break (%)
PLA/PBAT	243.1 ± 18.1	13.9 ± 0.2	132.7 ± 8.0
PLA-g-MAH_0.5	352.0 ± 4.3	17.3 ± 0.3	113.1 ± 5.4
PLA-g-MAH_1.0	392.7 ± 30.4	17.4 ± 0.5	327.8 ± 18.3
PLA-g-MAH_1.5	423.7 ± 25.8	16.8 ± 0.2	316.7 ± 8.8
PBAT-g-MAH_0.5	521.6 ± 20.9	21.8 ± 1.6	289.8 ± 9.7
PBAT-g-MAH_1.0	590.4 ±17.1	24.8 ± 0.6	325.3 ± 4.7
PBAT-g-MAH_1.5	597.5 ± 8.2	22.7 ± 0.8	311.3 ± 14.6
(PLA/PBAT)-g-MAH_0.5	644.3 ± 13.3	25.3 ± 0.4	256.7 ± 9.4
(PLA/PBAT)-g-MAH_1.0	679.3 ± 23.4	27.7 ± 0.7	380.9 ± 3.1
(PLA/PBAT)-g-MAH_1.5	589.7 ± 41.9	22.7 ± 2.8	311.8 ± 10.8

## Data Availability

The original contributions presented in the study are included in the article, further inquiries can be directed to the corresponding author.

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
