# Peer review of "Enhanced Mechanical Properties of Polylactic Acid/Poly(Butylene Adipate-co-Terephthalate) Modified with Maleic Anhydride"

_polymers, 2024, doi:10.3390/polym16040518_

Round 1

Reviewer 1 Report

Comments and Suggestions for Authors

Dear Authors,

Overall, this is an interesting manuscript. I also think that the authors definitely put a lot of work into it. To improve its quality, I am including some important comments below.

The introduction is correctly written and well justifies the need to write this article. However, I have a few minor comments here.

At the beginning of the introduction you write about biodegradable materials from biomass. I think it is worth mentioning various examples of biocomposites made of such raw materials. It is also worth mentioning, for example, TPS. Browse the following papers: "Properties of biocomposites produced with thermoplastic starch and digestate: Physicochemical and mechanical characteristics" or "High-performance starch biocomposites with cellulose from waste biomass: Film properties and retrogradation behavior". Of course, there are more such works.

I also lack information about what new research this research brings to the development of PLA foil. You should therefore better justify the innovativeness of your research.

In the introduction, you use the term "plastics" and then write about the composition of plant raw materials, etc. Is this really correct? Check the entire text from this angle. Maybe it's worth using "biodegradable materials or biodegradable plastics, etc."

Methodology: The methodology describes the course of research well. Based on the information contained in the methodology, the research could be recreated. However, I have some minor comments.

1. Show clearly where the blowing process takes place.

2. Was the temperature of the extrusion process the same throughout the extruder cylinder - 180oC. If so, please provide the +/- deviation.

3. Describe the head speed when performing mechanical strength tests.

Results and Discussion. The description of the research results seems well done. However, the weakness of this article is the lack of comparison of the obtained test results with other materials. Overall, the discussion of the research results is somewhat poor. Few literature items. There could easily be 10 more of them in an article like this.

Conclusions: Many of these conclusions are simply statements. Add one more forward-looking conclusion.

Reviewer 2 Report

Comments and Suggestions for Authors

A quite interesting work on the combination of 2 types of polymers into an improved one. The work is, overall, well written and the English is good. Some minor observations.

1. Section 2.3. The last paragraph is about the tensile tests. The authors should not be so brief regarding a substantial point. For instance, even though the DMA test is described sufficiently, the tear test and the tensile tests are not equally presented. The specimen dimensions, especially the film thickness, the machine and measuring equipment used etc. are very important for readers that would like to perform similar tests.

2. Section 3.3. The results are well presented in terms of mechanical properties but not even 1 stress-strain curve or a load-displacement curve from the mechanical tests. The authors are strongly advised to add these curves to make more credible the results.

Round 2

Reviewer 1 Report

Comments and Suggestions for Authors

Dear Authors,

The authors have taken into account most of the corrections included in the previous review. Accepts all corrections.